# ⛏CARTS: Advancing Neural Theorem Proving with Diversified Tactic Calibration and Bias-Resistant Tree Search

**Xiao-Wen Yang**[1,2], **Zhi Zhou**[1], **Haiming Wang**[3,5], **Aoxue Li**[4]
**Wen-Da Wei**[1,2], **Hui Jin**[4], **Zhenguo Li**[4], **Yu-Feng Li**[1,2*]
[1]National Key Laboratory for Novel Software Technology, Nanjing University
[2]School of Artificial Intelligence, Nanjing University
[3]Sun Yat-sen University    [4]Noah's Ark Lab, Huawei    [5]Moonshot AI

## Abstract

Recent advancements in neural theorem proving integrate large language models with tree search algorithms like Monte Carlo Tree Search (MCTS), where the language model suggests tactics and the tree search finds the complete proof path. However, many tactics proposed by the language model converge to semantically or strategically similar, reducing diversity and increasing search costs by expanding redundant proof paths. This issue exacerbates as computation scales and more tactics are explored per state. Furthermore, the trained value function suffers from false negatives, label imbalance, and domain gaps due to biased data construction. To address these challenges, we propose CARTS (diversified tactic CAlibration and bias-Resistant Tree Search), which balances tactic diversity and importance while calibrating model confidence. CARTS also introduce preference modeling and an adjustment term related to the ratio of valid tactics to improve the bias-resistance of the value function. Experimental results demonstrate that CARTS consistently outperforms previous methods achieving a pass@1 rate of 49.6% on the miniF2F-test benchmark. Further analysis confirms that CARTS improves tactic diversity and leads to a more balanced tree search. The code for our implementation is available at `https://github.com/njuyxw/CARTS`.

## 1 Introduction

Automated theorem proving (ATP) (Harrison et al., 2014) is an essential task of artificial intelligence (AI) with significant challenge. Recently, the development of large language models has brought new vitality and advancements to this field (Han et al., 2022; Jiang et al., 2023; Xin et al., 2024a). For example, AlphaProof (Deepmind, 2024; Trinh et al., 2024) solved four out of six problems from International Mathematical Olympiad (IMO), achieving the same level as a silver medalist in the competition. These advancements stem from the integration of language models and formal theorem proving systems (such as Lean (Moura & Ullrich, 2021) or Isabella (Paulson, 1994)), which model the theorem proving task as a Markov Decision Process (MDP) (Polu & Sutskever, 2020). The language model functions as a policy network that provides heuristic proof tactics, while tree search methods are utilized to explore correct sequence of steps that maximize the reward.

Although the improvements of language models (Xin et al., 2024a) can significantly improve the performance of theorem proving, efficient tree search methods remains crucial for theorems with long and complex proof steps. Existing search techniques (Polu & Sutskever, 2020; Wang et al., 2023; Xin et al., 2024a) primarily rely on Best First Search (BFS) or Monte Carlo Tree Search (MCTS) (Kocsis & Szepesvári, 2006). While these method can achieve impressive performance, they have two significant drawbacks. **Firstly**, the output sampling of auto-regressive language models frequently exhibits significant redundancy, often producing similar tactics. Although the language model generates a substantial number of tactics that differ at the character level, they share the same underlying semantics. For instance, both 'intro h' and 'intro H' can be generated

---

*Corresponding author: Yu-Feng Li (`liyf@lamda.nju.edu.cn`)

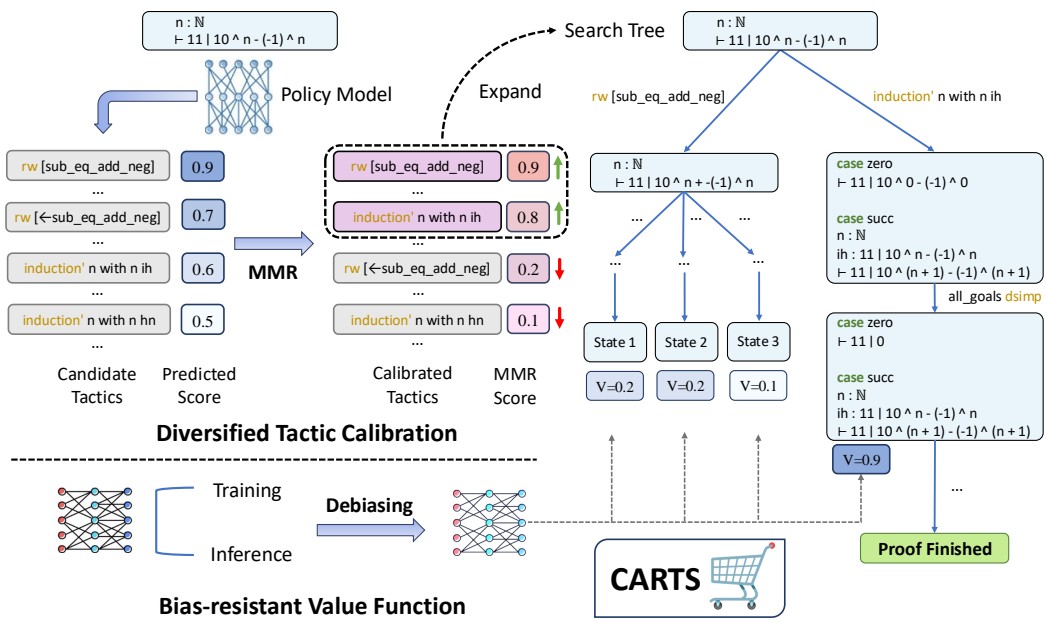

Figure 1: **Overall Framework.** Diversified tactic calibration can calibrate the model confidence and enhance the diversity of candidate tactics, thus mitigating ineffective exploration. The bias-resistant value function can adapt to the test data, provide more accurate scores for evaluating tactics, thus improving the efficiency of utilization.

by the language model; however, they convey the same meaning in Lean4, as they both introduce a hypothesis into the proving context. This redundancy can also lead to an imbalance in the number of high-level proof strategies. For example, most of the tactics generated by the language model may focus on the strategy of proof by contradiction, while there are relatively few tactics involving mathematical induction. These will result in a huge amount of ineffective exploration during the tree search process, thus increasing search costs. This issue worsens as computation scales and more tactics are explored per state. **Secondly**, the construction of value function training data often relies on existing policy models to generate negative samples and thus introducing bias. On one hand, this construction may produce a substantial number of negative samples, potentially far exceeding the positive ones. This could result in label imbalance, causing the cross-entropy loss used in training the value function to easily converge to local optima. On the other hand, the samples generated by the policy model may contain false negatives, introducing noise into the dataset. Additionally, the domain gap between the training dataset (e.g., Mathlib) and the test dataset (e.g., IMO problems) exacerbates the bias of the value function during the inference stage. This leads to inaccurate evaluations of the current proof state's value, hindering effective exploitation during the tree search process. Overall, these two issues prevent existing tree search techniques from efficient exploration and effective exploitation, resulting in sub-optimal search performance.

In order to solve these challenge, we propose diversified tactic **CA**libration and bias-**R**esistant **T**ree **S**earch (**CARTS**). Diversified tactic calibration involves reordering and rescoring multiple candidate tactics generated by a language model's sampling output. This approach balances importance and diversity, thereby enhancing exploration efficiency. We use the Maximal Marginal Relevance (MMR) algorithm (Peng et al., 2005) to achieve this, which is a classical method in the field of information retrieval. Meanwhile, we propose a bias-resistant value function. During the training stage, preference modeling is employed to construct the training dataset, and the Bradley-Terry model (Bradley & Terry, 1952) is utilized to train the value network. This approach addresses the issue of data imbalance and false negatives. During the inference stage, we introduce an adjustment term related to the ratio of valid tactics into the value function to mitigate the domain gap between the training and test dataset. This stems from a insight that if the number of valid tactics is limited, concerns

may arise regarding the effectiveness of the current policy model, necessitating a reduced value for the current action. Bias-resistant value function can enhance the effectiveness of exploitation during the search process. The complete framework of CARTS is shown in Figure 1. We conducted sufficient experiments on the widely recognized theorem-proving benchmarks, namely miniF2F (Zheng et al., 2022) and ProofNet (Azerbayev et al., 2023) in Lean. Our proposed CARTS demonstrates superior performances compared to all other search methods when the policy network remains unchanged. We achieved a pass@1 success rate of 49.6% on the miniF2F-test dataset, which is the state-of-the-art performance among all one-step tree search methods.

To summarize, this paper (i). proposes a diversified tactic calibration assisted monte carlo tree search to improve the exploration efficiency. (ii). proposes a bias-resistant value function to improve the exploitation effectiveness. (iii). demonstrates the effectiveness of the proposed method across different models and benchmarks in experiments.

## 2 RELATED WORK

**Neural theorem proving.** In recent years, the advancement of large language models has brought new progress to theorem proving (Li et al., 2024). GPT-f (Polu & Sutskever, 2020) is the first to utilize language models trained on proof data to predict candidate proof steps and employ search algorithms to discover the complete proof path. A series of subsequent studies have employed diverse language model techniques from various perspectives to enhance theorem proving performance. In terms of model training, PACT (Han et al., 2022) employs a set of self-supervised auxiliary tasks to train the model. Curriculum Learning (Polu et al., 2023) introduces curriculum expert iterations to update the network. Llemma (Azerbayev et al., 2024) continues pre-training the CodeLlama models (Roziere et al., 2023) on a math-focused corpus. AlphaGeometry (Trinh et al., 2024) integrates a transformer model trained on synthetic geometry data with a symbolic deduction engine to solve olympiad geometry problems. InterLM2-Math (Ying et al., 2024b) compiles a substantial collection of both formal and informal contest-level math problems (Ying et al., 2024a; Wu et al., 2024). First et al. (2023) incorporates a repair feedback mechanism in proof generation. This feedback is facilitated by an LLM fine-tuned on tuples consisting of incorrect proof, error message and correct proof. In terms of algorithmic design, Reprover (Yang et al., 2023) employs retrieval-augmented generation for proof generation. DSP (Jiang et al., 2023) initially uses informal hints to guide proofs by translating informal proofs into formal sketches, which are then completed with Isabelle's automated reasoning tactics. LEGOProver (Xin et al., 2024c) enhances DSP with a skill library that expands throughout the proof search. Lyra (Zheng et al., 2024) iterates on DSP by using error feedback to modify the formal sketch, employing automated reasoning tools to correct incorrect proofs of intermediate hypotheses. COPRA (Thakur et al., 2024) utilizes in-context learning agents to augment theorem proving. These methods employ different formal systems. In our paper, we focus on Lean, which has been verified to perform well on IMO-level tasks (Deepmind, 2024).

**Search methods for theorem proving.** Neural theorem proving primarily consists of two categories: whole proof generation methods Xin et al. (2024a); Wang et al. (2024b) and tree search methods. Tree search methods are increasingly becoming the mainstream approach in recent years. A typical approach involves using Best First Search (BFS), as seen in methods like GPT-f (Polu & Sutskever, 2020), Reprover (Yang et al., 2023) and others (Lin et al., 2024; Welleck & Saha, 2023). In contrast, Thakur et al. (2024) employs depth-first search (DFS). Inspired by AlphaZero (Silver et al., 2018), many methods utilize MCTS, such as HyperTree Proof Search (Lample et al., 2022). There are also several improvements to the MCTS algorithm for theorem proving tasks. For instance, DT-Solver (Wang et al., 2023) uses virtual nodes and a proof-level value function to dynamically guide the MCTS search. Wang et al. (2024a) introduces a novel method that allows for the emergence of unproven lemmas during the search, which are subsequently proven recursively. The aforementioned methods are all one-step tree search techniques, which generates a single tactic at each step. Recently, multi-step tree search methods have been developed. For instance, DeepSeek-Prover-V1.5 (Xin et al., 2024b) employs MCTS to enhance the whole proof generation process, utilizing intrinsic rewards and discounted upper confidence bounds to guide exploration. Despite the success of these methods, challenges remain, namely the lack of diversity for searched proof paths and bias in the trained value function. This paper focuses on addressing these challenges by employing our diversified tactic calibration and bias-resistant tree search.

---

**Algorithm 1** Maximal Marginal Relevance for Diversified Tactic Calibration

---

**Require:** current state $s$, tactics set $\{a_1, a_2, ..., a_e\}$, next state set $\{s'_1, s'_2, ..., s'_e\}$, number of selected tactics $k$, parameter $\lambda$

    $S \leftarrow \{s\}$
    $A \leftarrow \{\}$
    **while** $|A| < \min(k, e)$ **do**
        $a^* \leftarrow \arg\max_{a_i} \left( \lambda \cdot v_{policy}(s, a_i) - (1 - \lambda) \cdot \max_{s'_j \in S} f_{enc}(s'_i)^\top f_{enc}(s'_j) \right)$
        Add $a^*$ to $A$
        Add the corresponding next state $s^*$ to $S$
    **end while**
    **return** the expanded action set $A$

---

## 3 METHOD

In this section, we present the details of our proposed method CARTS, which consists of two components. We begin by introducing the diversified tactic calibration (3.1), followed by giving details of our bias-resistant value function (3.2). Together, these two approaches enhance the effectiveness of exploration and exploitation during the search process.

### 3.1 DIVERSIFIED TACTIC CALIBRATION

In practice, the multiple candidate tactics generated by language models often exhibit redundancy. Diversified tactic calibration addresses this by calibrating candidate tactics' model confidence based on their intrinsic similarity. We implement the calibration using the Maximal Marginal Relevance (MMR) algorithm (Peng et al., 2005) and structure our method's framework through Monte Carlo Tree Search (Kocsis & Szepesvári, 2006).

The standard MCTS method used in theorem proving (Wang et al., 2023; Xin et al., 2024b) involves three steps: *Selection*, *Expansion* and *Backpropagation*. We incorporate diversified tactic calibration into the *Expansion* phase, resulting in our CARTS method, which comprises three steps: *Selection*, *Calibration & Expansion*, and *Backpropagation*. Details of each step are provided as follows.

**Selection.** In the selection phase, the algorithm starts from the root node and traverses the tree down to a leaf node. It uses a tree policy to choose child nodes that balance exploration and exploitation. The tree policy at a tree node $s$ selects an action $a$ that maximizes the weighted upper confidence bound (WUCB) score, the WUCB score for each tree node $s$ is formulated as follows:

$$\text{WUCB}(s, a) = \frac{W(s, a)}{N(s, a)} + w(s, a) \cdot \frac{\sqrt{N(s, \cdot)}}{N(s, a)} \tag{1}$$

Here, $N(s, a)$ denotes the count of how many times action $a$ has been taken in state $s$ and $N(s, \cdot)$ the total number of times any action has been taken in state $s$ during the whole search. $W(s, a)$ denotes the total value accumulated. Unlike the PUCT score used in DT-Solver (Wang et al., 2023), which incorporates probabilities estimated by the language model, we introduce a weight $w(s, a)$ that represents both importance (model confidence) and diversity for the tactic at the current state. We will detail the weights in the calibration phase.

**Calibration & Expansion.** At this stage, multiple candidate tactics are generated from the language model, followed by verification through the Lean prover. Verified tactics that pass in Lean are calibrated and expanded into the search tree. Concretely, the proof generation model is designed to generate a one-step proof tactic $a$ from a given proof state $s$, along with a conditional probability $p(a|s)$. Typically, we use beam search to sample a large collection of tactics (the quantity is $E$) from the language model, which may result in much low probabilities for each tactic. This will lead to reduced exploration during the search process. To fix this, we apply a length penalty, defined as $v_{policy}(s, a) = p^{\frac{1}{l}}(s|a)$, where $l$ is the token length of tactic $a$. This value reflects the model's confidence or the importance of the current action.

After verification by Lean, only $e$ tactics remain, denoted as $\{a_1, a_2, ..., a_e\}$. Here, we do not directly expand these tactics into the search tree because they often exhibit significant redundancy. Therefore, we need to capture the similarity between these tactics and then reorder them for both diversity and importance. We can compare the similarities between two actions using a pre-trained and fixed sentence encoder. However, directly using action similarity poses a challenge: similar actions may lead to different next states, or dissimilar actions may result in the same next state. This undermines our goal of enhancing search diversity. Therefore, we assess next states similarities after executing these tactics. Formally, we denote the set of the next states as $\{s_1', s_2', \ldots, s_e'\}$, where $s_i'$ represents the next state of $s$ following the execution of action $a_i$. We use a sentence encoder $f_{enc}(\cdot)$, which has been already pre-trained on a large-scale corpus, to accept the textualized state and outputs high-dimensional embeddings. Then we utilize the MMR algorithm to reorder the tactics and calibrate the model confidence. The algorithm iteratively selects items (e.g., tactics in our context) from a candidate set to maximize the following objective function:

$$\text{MMR}(s, a_i) = \lambda \cdot v_{policy}(s, a_i) - (1 - \lambda) \cdot \max_{s_j' \in S} f_{enc}(s_i')^\top f_{enc}(s_j') \tag{2}$$

Where, $\lambda$ is a parameter that controls the trade-off between importance and diversity. It typically ranges from 0 to 1. The calculation of the MMR score serves a calibration, effectively penalizing tactics with low diversity, as represented by the second term in the formula. The algorithm begins with an initial state set $S = \{s\}$ and an empty action set $A$. While the size of $A$ is less than the predefined value $k$, which is smaller than $E$, the action with the highest MMR score is selected and added to $A$, along with its next state being added to $S$. Once $k$ actions have been chosen, the set $A$ is returned. If $k < e$, we select all the tactics and reorder them. The value $k$ serves as a constraint on the maximum number of expansion nodes. It is worth noting that we add the current state $s$ into the initial state set $S$ to mitigate the recurrence of identical states. The reason this way is effective is that our algorithm assigns a lower MMR score to the actions when the next state closely resembles the current state. Algorithm 1 illustrates the complete process of diversified tactic calibration.

After diversified tactic calibration, we obtain a small set of actions $A$ that contains both importance and diversity. Then, these actions are treated as edges, with their corresponding next state as nodes, which are expanded into the current search tree. For each edge $a_i$, we assign a weight $w(s, a_i) = \max\{0, \text{MMR}(s, a_i)\}$, utilized during the selection phase to assess the need for exploration. Unlike traditional MCTS (Kocsis & Szepesvári, 2006) or DT-Solver (Wang et al., 2023), our weights places greater emphasis on encouraging the exploration of tactics with high diversity.

**Backpropagation.**  At this stage, we update the statistics of the nodes and edges along the search trajectory. We have a bias-resistant value function $V(s, a)$ which will be detailed in the next section, estimating the value of taking action $a$ from the source node $s$. For a given trajectory, we use the value function to evaluate the value of the leaf node and accumulate this value along all edges in the path. Specifically, we update the weight of the edge recursively as follows: $W(s_t, a_t) \mathrel{+}= V(s, a)$, where $s_t$ and $a_t$ represent the node and edge at the trajectory. Additionally, we increment the visit count for the edge: $N(s_t, a_t) \mathrel{+}= 1$. This process ensures that the statistics reflect the outcomes of the simulations, allowing for improved selection in future iterations.

## 3.2 Bias-resistant Value Function

In MCTS-based methods, training a value function is crucial (Polu & Sutskever, 2020; Lample et al., 2022), typically involving the creation of positive and negative samples using the policy network on training data. Positive samples consist of correct actions (or trajectories (Wang et al., 2023)) from the dataset, while negative samples are those generated by the policy network that lead to undesirable states. Binary cross-entropy loss is then used to train the value network. Due to the hardness of verifying the correctness of actions not on the proof path, previous work (Polu & Sutskever, 2020) often treats these actions as negative samples, resulting in an excessive number of negative samples, some of which are even inaccurate. This makes binary loss unsuitable and biases the value function. Furthermore, the domain gap between the training and test datasets also contributes to biases. In this paper, we conduct debiasing during both training and inference stages, as detailed below.

**Training.**  To mitigate bias introduced by data collection, we first structure the dataset into preference pairs of positive and negative samples. We utilize an embedding model $f_{enc}$ to effectively

filter out noisy samples. Specifically, if $f_{enc}(s')^\top f_{enc}(s'_{pos}) > \tau$, we discard the action $a$. $s'$ is the next state from a sampled negative action $a$ at the current state $s$, $s'_{pos}$ is the correct next state and $\tau$ is a threshold. This filtering ensures that the selected negative actions are more likely to be undesirable, thus reducing data noise. Moreover, we adopt the preference modeling framework to train our bias-resistant value function. We employ the Bradley-Terry (BT) (Bradley & Terry, 1952) model, a widely used technique for preference modeling. The BT model posits that the probability of action $a_{pos}$ being preferred over action $a_{neg}$ given state $s$ is expressed as:

$$\mathbb{P}(a_{pos} \succ a_{neg} \mid s) = \frac{\exp(V_\theta(s, a_{pos}))}{\exp(V_\theta(s, a_{pos})) + \exp(V_\theta(s, a_{neg}))} \quad (3)$$

Assuming access to the filtered dataset $D = \{(s^{(i)}, a_{pos}^{(i)}, a_{neg}^{(i)})\}_{i=1}^N$, we can parametrize the value function $V_\theta(s, a)$ and estimate the parameters $\theta$ by minimizing the negative log-likelihood.

Preference modeling offers several advantages. Firstly, by only providing the relative superiority among samples, false negative samples do not require further processing. This is because we can reasonably assume that the correct proof steps provided in the dataset are always optimal and align with human theorem proving's preferences. Additionally, since the dataset is presented in the form of preference pairs, this effectively oversamples (Shi et al., 2023) the positive pairs, alleviating the issue of class imbalance between positive and negative samples, as demonstrated in some studies (Zhang et al., 2024; Pattnaik et al., 2024).

**Inference.** To mitigate the domain gap between the training and test datasets, we introduce an adjustment term into the value function during the inference stage. As previously mentioned, before calibration in CARTS, all $E$ tactics should be processed through the Lean system to filter out $e$ valid tactics. Intuitively, if the number of valid tactics is small, people will raise concerns about the capability of the current policy model, needing for a reduced reward for the current action. We define this reward adjustment as: $\alpha = e/E$, representing the ratio between the number of valid tactics and the total number of tactics generated by the language model at the current state. This adjustment term serves as a test-time adaptation to the test dataset. The final bias-resistant value function integrates both the trained value network and this adjustment term as:

$$V(s, a) = \begin{cases} 0, & \text{if } s' \text{ has no child nodes,} \\ 1, & \text{else if } s' \text{ is the proved state,} \\ \frac{1}{2}(\alpha + V_\theta(s, a)), & \text{otherwise.} \end{cases} \quad (4)$$

Where $s'$ is the next state. Unlike the intrinsic reward introduced by DeepSeek-Prover-V1.5 (Xin et al., 2024b), which only considers whether the search expands nodes, we consider both the expansion capability of the policy network and the generalizability of the value network, forming our final bias-resistant value function. The adjustment term can be interpreted as a form of test-time adaptation to the distribution of test data, thus can mitigate the domain gap.

## 4 EXPERIMENTS

In this section, we evaluate the theorem-proving performance of CARTS in Lean. We first describe the experimental setup, then present the main results, followed by an analysis of our method. Currently, theorem-proving methods are primarily categorized into two main types: whole-proof generation methods and tree search methods. Our approach is applicable exclusively to one-step tree search methods; therefore, we focus our comparison solely on this category.

### 4.1 EXPERIMENTAL SETUP

**Datasets.** We follow Internlm-math (Ying et al., 2024b; Wu et al., 2024) and DeepSeek-Prover-V1.5 (Xin et al., 2024b), utilizing miniF2F benchmark (Zheng et al., 2022) and ProofNet benchmark (Azerbayev et al., 2023) for our evaluation. We specifically use the test split of miniF2F same as (Xin et al., 2024b), which includes 244 problems ranging from basic algebra and number theory and also contains AIME and IMO challenging problems. ProofNet is a benchmark for undergraduate-level mathematics, comprising 371 formal problems derived from widely-used undergraduate pure mathematics textbooks. It covers topics such as real and complex analysis, abstract algebra, and

Table 1: Results on the miniF2F-test for various models and search methods. The highest performance for each search method is highlighted in **bold**.

| Model | Sample Budget | Search Method | miniF2F-test |
|---|---|---|---|
| *Tree Search Methods* | | | |
| COPRA(Code Llama) (Thakur et al., 2024) | 500 | DFS | 5.7% |
| COPRA(GPT-3.5) (Thakur et al., 2024) | 60 | DFS | 9.0% |
| COPRA(GPT-4) (Thakur et al., 2024) | 60 | DFS | 26.6% |
| Llemma-7B (Azerbayev et al., 2024) | $32 \times 100$ | BFS | 26.2% |
| Llemma-34B (Azerbayev et al., 2024) | $32 \times 100$ | BFS | 25.8% |
| LLMStep (Welleck & Saha, 2023) | $32 \times 100$ | BFS | 27.9% |
| Curriculum Learning (Polu et al., 2023) | $8 \times 512$ | BFS | 29.6% |
| InternLM2-Math-7B (Ying et al., 2024b) | $32 \times 100$ | BFS | 30.3% |
| InternLM2-Math-Plus-7B (Ying et al., 2024a) | $32 \times 100$ | BFS | 43.4% |
| DeepSeek-Prover-V1.5-SFT (Xin et al., 2024b) | 3200 | RMaxTS | 53.5% |
| DeepSeek-Prover-V1.5-RL (Xin et al., 2024b) | 3200 | RMaxTS | 55.0% |
| Reprover-Lean4 (229M) (Yang et al., 2023) | $64 \times 100$ | BFS | 35.7% |
| | | MCTS | 36.5% |
| | | DTSolver | 36.0% |
| | | CARTS | **37.7%** |
| InternLM2-Math-Plus-1.8B (Ying et al., 2024b) | $64 \times 100$ | BFS | 38.9% |
| | | MCTS | 39.3% |
| | | DTSolver | 38.5% |
| | | CARTS | **41.0%** |
| StepProver (7B) (Wu et al., 2024) | $32 \times 300$ | BFS | 48.8% |
| | | MCTS | 46.7% |
| | | DTSolver | 46.3% |
| | | CARTS | **49.6%** |

topology. This benchmark presents a greater challenge than miniF2F, posing significant difficulties for theorem provers. Although the original versions of both benchmarks are Lean3, we have modified them to Lean4 for CARTS's evaluation, aligning with the development of the Lean community.

**Baselines models.** We include baselines representing classical and state-of-the-art neural theorem proving in Lean. **COPRA** (Thakur et al., 2024) is an in-context learning agent that utilizes general language models to generate tactics for finding the final proof. **Llemma** (Azerbayev et al., 2024) is trained on extensive mathematical corpora. Additionally, we incorporate advanced models such as **LLMStep** (Welleck & Saha, 2023), **Reprover** (Yang et al., 2023), **Curriculum Learning** (Polu et al., 2023), **InternLM2-Math** (Ying et al., 2024b), and **StepProver** (Wu et al., 2024). All these models are based on one-step tree search methods. We also include **DeepSeek-Prover-V1.5** (Xin et al., 2024b), which integrates whole proof generation and tree search. However, it is not suitable for our CARTS, and thus, we mark it in gray.

**Search methods.** Baseline models employ various search methods, such as depth-first search (**DFS**) (Thakur et al., 2024), best first search (**BFS**) (Yang et al., 2023), and monte carlo tree search (**MCTS**). Additionally, **DT-Solver** extends MCTS using virtual nodes. We compared the performance of BFS, MCTS, DT-Solver and CARTS on Reprover (Yang et al., 2023), InternLM2-Math (Ying et al., 2024b), and InternLM2-StepProver (Wu et al., 2024). For MCTS and DT-Solver, we replace the value network with the intrinsic reward (Xin et al., 2024b) for simplication. It is noteworthy that multiple tree search attempts with different seeds can be applied and ensemble (Polu & Sutskever, 2020; Lin et al., 2024; Xin et al., 2024a); however, due to computational cost limitation, we only compared the results for one single tree search attempt.

**Metrics.** We evaluate the performance of various search methods using the pass@1 metric with a budget $B$. Similar to (Xin et al., 2024b), if $B$ is a single value, it indicates the number of model

Table 2: Results on the ProofNet for various models and search methods.

| Model | Sample Budget | Search Method | ProofNet |
|---|---|---|---|
| Reprover-Lean4 (Yang et al., 2023) | $64 \times 100$ | BFS | 11.1% |
| | | MCTS | 11.7% |
| | | CARTS | **11.9%** |
| StepProver (Wu et al., 2024) | $32 \times 300$ | BFS | 18.1% |
| | | MCTS | 18.3% |
| | | CARTS | **18.8%** |

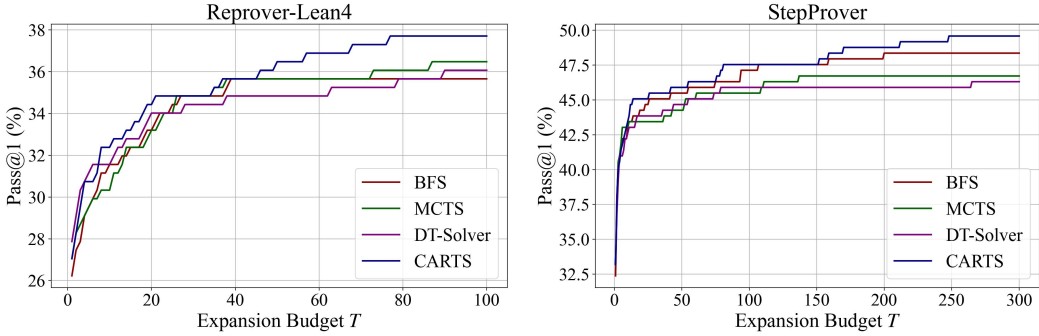

Figure 2: Improvement curve in pass@1 for the miniF2F test as the expansion budget varies. Left illustrates Reprover-Lean4 (Yang et al., 2023) and right illustrates StepProver (Xin et al., 2024b).

generations used in tree expansions. If $B = E \times T$, $E$ represents the number of tactics generated per expansion, and $T$ denotes the number of expansion iterations.

**Experimental details.** In terms of parameter settings, for Reprover-Lean4 (Yang et al., 2023), we set $\lambda = 0.8$, and for InternLM2-Math-Plus-1.8B (Ying et al., 2024b) and StepProver (Wu et al., 2024), we set $\lambda = 0.9$. Additionally, we set $k = 8$ for all models. We use the text embedding model `intfloat/e5-small-v2` (Wang et al., 2022) as the encoder $f_{enc}$. Details regarding data collection and training for the bias-resistant value function are presented in Appendix A.

## 4.2 MAIN RESULTS

In Table 1, we illustrate the pass@1 successful rate on the miniF2F-test benchmark of various models and search methods. Our proposed CARTS surpasses all compared search methods when the policy model is fixed. The results encompass different architectures and parameter sizes of language models, demonstrating that our method is more effective at search stage regardless of the policy model. Notably, we achieve a 49.6% success rate on the miniF2F-test, representing state-of-the-art performance among one-step tree search methods. Table 2 demonstrates the results on the ProofNet benchmark. Due to the dataset's complexity, current policy models exhibit low accuracy, which constrain the search performances. However, our proposed CARTS also achieves the highest performance compared to other search methods when the budget is fixed.

Additionally, to demonstrate that CARTS has the superior search efficiency, we present a comparison of the pass@1 performance among four search methods on miniF2F-test when varying the expansion budget, as illustrated in Figure 2. It is evident that while the performance of different search methods is comparable under a low budget, our method exhibits a significant improvement as the expansion budget increases. Furthermore, we observed that the CARTS curve increases more rapidly relative to other search methods. These findings demonstrate that our approach enhances the efficiency and effectiveness of tree search.

Table 3: Ablation study of various components of CARTS on the miniF2F-test. We compare the pass@1 performance across three models, with the highest performance highlighted in bold.

| Search Method | Model | | |
|---|---|---|---|
| | Reprover-Lean4 | InternLM2-Math-Plus-1.8B | StepProver |
| ▷ *Baselines:* | | | |
| BFS | 35.7% | 38.9% | 48.8% |
| MCTS | 36.5% | 39.3% | 46.7% |
| ▷ *Ablations:* | | | |
| Diversified tactic calibration | 36.1% | 41.0% | 49.2% |
| Bias-resistant value function | 36.9% | 40.6% | 46.3% |
| CARTS (without adjustment term) | 37.3% | **41.3%** | 49.2% |
| CARTS (Ours) | **37.7%** | 41.0% | **49.6%** |

## 4.3 ANALYSIS

In this section, we conduct a detailed analysis of CARTS' effectiveness through experiments. We include ablation studies and specifically investigate the length distribution of results produced by CARTS, supplemented by several case studies.

**Ablation for different components of CARTS.** CARTS consists of two components: diversified tactic calibration and a bias-resistant value function. We isolated these components for analysis. Additionally, we compared the performance of CARTS without the adjustment term in the value function. The experimental results are shown in Table 3. The results indicate that both components enhance performance, and their combination further increases the pass rate, suggesting that CARTS can significantly improve the efficiency of tree search exploration. Our findings reveal that for InternLM2-Math-Plus-1.8B, CARTS without the adjustment term achieves superior results, whereas CARTS with the adjustment does not perform optimally. We believe this is because the adjustment term is merely a heuristic calibration; the policy model generates more unacceptable tactics, the less likely it is to generate accurate tactics. This is not always valid due to differences between the properties of the proof state. Nonetheless, experiments across multiple models suggest that it is generally effective for debiasing the value function.

**Ablation study for $k$.** The parameter $k$ limits the maximum expansion number of nodes after diversified tactic calibration. A smaller $k$ significantly restricts the search space, potentially leading to performance degradation. Conversely, a larger $k$ expands the search space, increasing the likelihood of identifying the correct tactic. However, this comes at the cost of substantially increased search time, thereby reducing overall efficiency. To assess its impact on CARTS, we test multiple values of $k$ using Reprover-Lean4 on the miniF2F-test. The experimental results are shown in the left side of Figure 3. As $k$ increases, the pass@1 rate initially improves, reaching a peak, and then gradually declines. This indicates that there is an optimal range for $k$ that balances performance and search efficiency. Therefore, we recommend choosing a moderate $k$, such as $k = 8$, in practice.

**Analysis of the length distribution of discovered proofs.** To comprehensively demonstrate the effectiveness of CARTS, we analyze the proof step lengths obtained by the Reprover-Lean4 model on the miniF2F-test. The results are presented on the right side of Figure 3. As illustrated, BFS yields a higher proportion of theorems with step lengths of one and two, but with none exceeding three steps. In contrast, CARTS achieves more theorems with lengths of three or greater, significantly outperforming both BFS and MCTS. This indicates that CARTS enhances search efficiency, enabling the exploration of longer proof chains within the same budget. Additionally, we find that CARTS discover two proofs with the length of five. We examine this case in detail in next paragraph.

**Case study.** We conduct a comprehensive case study to closely examine the results of the CARTS search. Figure 4 demonstrates one theorem which is proved by CARTS on Reprover-Lean4 model.

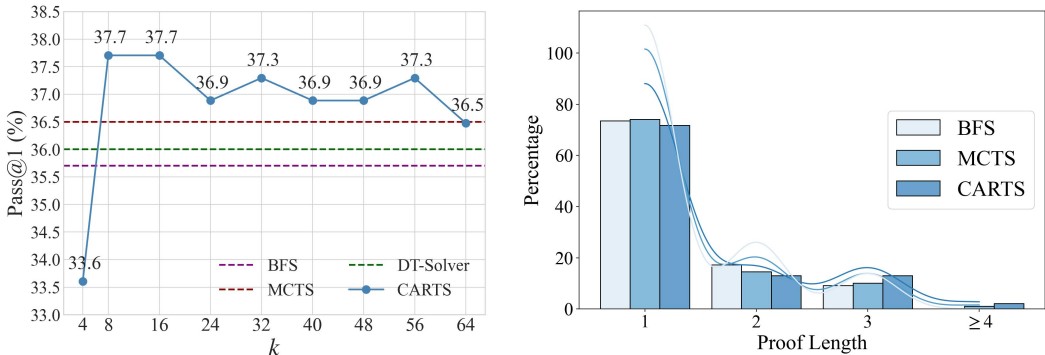

Figure 3: The left figure shows the ablation study for $k$, while the right figure illustrates the length distribution of proofs searched by CARTS. Both analyses are conducted on the miniF2F-test using the Reprover-Lean4 (Yang et al., 2023) model.

---

Case: `induction_12dvd4expnp1p20`

```
theorem induction_12dvd4expnp1p20 (n : ℕ) : 12 | 4^(n +1) +20 :=
by
    norm_num [Nat.pow_succ]
    induction' n with n n_ih
    case zero => simpa using n_ih
    rw [pow_succ]
    omega
```

---

Figure 4: **Case study.** Proved theorem `induction_12dvd4expnp1p20` by CARTS in mini-F2F-test of ReProver-Lean 4 (Yang et al., 2023). This theorem is proved in 77 expansions.

This case has proof path of five length. This theorem `induction_12dvd4expnp1p20` pertains to divisibility and its initial step involves simplification using `norm_num [Nat.pow_succ]`. However, the tactic generated by Repover-Lean4 exhibits a low confidence level, only ranking eight compared to other incorrect tactics. If the BFS approach is to be applied, it would necessitate the exploration of numerous redundant tactics before arriving at this specific action. Our experiments reveal that the CARTS improves the score of this tactic and increases its rank to six. Furthermore, we find that the score of the second tactic, namely `induction' n with n n_ih`, improves from an initial rank of sixth to fourth. While the increase in the ranking of the correct tactic score is only by two, the exponential growth in the complexity of tree search highlights the significance of early exploration of the correct action at the root node. Therefore, these improvements in rankings play a crucial role in successfully proving this theorem within a limited number of expansion.

## 5    CONCLUSION AND LIMITATION

In this paper, we presented CARTS, a novel approach that combines diversified tactic calibration with a bias-resistant value function for enhancing the performance of neural theorem proving. Our diversified tactic calibration effectively mitigates the redundancy in language model generations, promoting a more diverse and balanced exploration of candidate tactics. The bias-resistant value function addresses the prevalent biases during both the training and inference stage, thereby refining the exploitation capabilities of the tree search by providing more accurate and reliable evaluations of tactics. Experimental results on miniF2F and ProofNet have demonstrated that CARTS significantly outperforms existing one-step tree search methods, achieving state-of-the-art results.

While CARTS has proven effective, it also has limitations. Our method currently only supports one-step tree search, whereas multi-step tree search, such as DeepSeek-Prover-V1.5 (Xin et al., 2024b), has shown promise but is not supported by our approach. In the future, we will explore variants of CARTS that can be integrated into multi-step tree search methods.

ACKNOWLEDGMENT

This research was supported by the Key Program of Jiangsu Science Foundation (BK20243012), Leading-edge Technology Program of Jiangsu Science Foundation (BK20232003), the Jiangsu Science Foundation (BG2024036) and the Fundamental Research Funds for the Central Universities (022114380023). We would like to thank the reviewers for their constructive suggestions.

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

# A    DETAILS OF BIAS-RESISTANT VALUE FUNCTION

**Data Construction**    To train our bias-resistant value function on Lean4, we utilized two data sources. The first source is the random portion of the `leandojo_benchmark` (Yang et al., 2023), which is derived from mathlib4 (commit: `3ce43c18f614b76e161f911b75a3e1ef641620ff`). The second source is `Lean-Github` (Wu et al., 2024), a recently released dataset compiled from over 100 Lean 4 repositories. We employed InternLM2-Math-Plus-7B (Ying et al., 2024b) to generate eight candidate tactics for each proof state in the dataset. Tactics that differed from the correct tactic in the generations are used as negative samples, which we then paired with positive samples to form preference pairs. However, as discussed in the main text, this dataset contains significant noise, with positive and negative samples being semantically similar, as illustrated in Figure 5. To address this, we filtered the samples based on their similarity, using the embedding model `intfloat/e5-small-v2` (Wang et al., 2022) and a threshold $\tau = 0.85$. After filtering, the dataset consists of **1,797,951** (about 1800K) training preference pairs and **2,000** test preference pairs.

```
STATE:                                    STATE:
case a.h                                  case a.h
m : Type u                                m : Type u
n : Type v                                n : Type v
α : Type w                                α : Type w
inst+⁴ : DecidableEq n                    inst+⁴ : DecidableEq n
inst+³ : Fintype n                        inst+³ : Fintype n
inst+² : DecidableEq m                    inst+² : DecidableEq m
inst+¹ : Fintype m                        inst+¹ : Fintype m
inst+ : CommRing α                        inst+ : CommRing α
A : Matrix n n α                          A : Matrix n n α
i j : n                                   i j : n
⊢ ∀ (x : Perm n),                         ⊢ ∀ (x : Perm n),
    x ∈ univ →                                x ∈ univ →
      ↑↑(↑sign x) * ∏ i_1 : n, updateRow A I     ↑↑(↑sign x) * ∏ i_1 : n, updateRow A I
(Pi.single j 1) (↑x i_1) i_1 =            (Pi.single j 1) (↑x i_1) i_1 =
      ↑↑(↑sign x) * ∏ i_1 : n, updateColumn A     ↑↑(↑sign x) * ∏ i_1 : n, updateColumn A
j (Pi.single i 1) (↑x i_1) i_1           j (Pi.single i 1) (↑x i_1) i_1

POSITIVE ACTION:                          POSITIVE ACTION:
intro σ _                                 intro σ _
NEGATIVE ACTION:                          NEGATIVE ACTION:
intros σ _                                simp only [updateRow_apply, updateColumn_apply]
```

**Before Filtering**                          **After Filtering**

Figure 5: This illustrates an example of our dataset organization: the left side contains unfiltered data with noise, while the right side shows the samples after filtering.

**Training Details**    We utilized the `microsoft/phi-1.5` model (Li et al., 2023), which contains 1.3 billion parameters. Phi-1.5 has been trained on a large-scale code corpus, endowing it with a strong understanding of formal languages. We fine-tuned the model using LoRA (Hu et al., 2021) for one training epoch, using the learning rate as $1 \times 10^{-4}$. After training, the model attained an a ccuracy of **94.10%** on the test dataset. The training loss curve and validation accuracy curve are illustrated in Figure 6.

# B    EFFECTIVENESS OF BIAS-RESISTANT VALUE FUNCTION

In this section, we supplement experiments to demonstrate the effectiveness of our bias-resistant value function and draw the following conclusions: 1. The Bradley-Terry (BT) model effectively addresses the issue of class imbalance in training the value function. 2. Our analysis underscores the critical importance of data filtering and provides a way for determining the hyperparameter $\tau$.

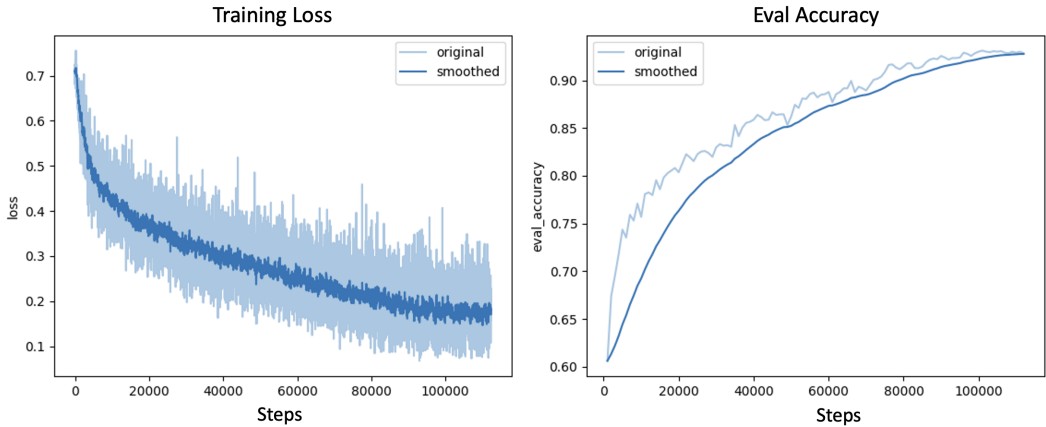

Figure 6: The training loss curve and the validation accuracy curve.

## B.1 ANALYSIS BETWEEN BT AND CE

We reconstruct the pairwise dataset and employ various models to validate the effectiveness of the BT modeling approach for the value function. Unlike the main text, where both `leandojo_benchmark` (Yang et al., 2023) and `Lean-Github` (Wu et al., 2024) are utilized as data sources, this analysis exclusively uses `leandojo_benchmark` for training, with its test split serving as the evaluation dataset. Furthermore, data frome `Lean-Github` are employed for Out-of-Distribution (OOD) evaluation. To enable training of Cross-Entropy (CE) loss, we convert the pair-wise datasets into the binary classification datasets. The training data exhibits an imbalance between positive and negative samples (approximately 1 : 5), while we ensured that both test sets remains class-balanced. Detailed statistics of the dataset are presented in Table 4.

Table 4: Statistics of our constructed dataset.

| Split | Count | |
|---|---|---|
| | **For BT** | **For CE** |
| Train | 694,600 | 861,186 |
| Test | 3,465 | 6,930 |
| Test-OOD | 3,499 | 6,998 |

We conducted training on three models to compare the differences between BT and CE, including Qwen-2.5-0.5B (Qwen Team, 2024), Llama3.2-1B (Dubey et al., 2024), and Llama3.2-3B (Dubey et al., 2024). We evaluated the accuracy (**Acc**) on `Test` and the out-of-distribution accuracy (**Acc-OOD**) on `Test-OOD`. All experiments are full fine-tuning for one epoch with a learning rate of 1e-5. The experimental results are presented in Table 5.

The experimental results show that BT significantly outperforms CE in terms of in-domain accuracy, highlighting the advantages of using BT for modeling. Furthermore, we observe a slight performance improvement in BT as model size increases; however, the improvements remains relatively small. We attribute this to the limited dataset size, which hinders the emergence of clear scaling properties. In addition, we observe that CE shows poor performance in scenarios with a significant domain gap, and its accuracy is almost the same as random guessing. In contrast, the BT model achieves approximately a 5% improvement in such cases, although its overall performance remains limited. We believe this is due to the significant domain gap between the `leandojo_benchmark` and `Lean-Github` datasets, making the task inherently challenging. (Note: In our main text, both datasets were included in the training set to mitigate the gap with miniF2F.)

Table 5: Results between BE and CE of different models.

| Model | Method | Acc (%) | Acc-OOD (%) |
|---|---|---|---|
| Qwen-2.5-0.5B (Qwen Team, 2024) | CE | 63.2 | 50.9 |
| | BT | **76.5** | **55.2** |
| Llama3.2-1B (Dubey et al., 2024) | CE | 67.4 | 52.2 |
| | BT | **76.6** | **54.6** |
| Llama3.2-3B (Dubey et al., 2024) | CE | 63.0 | 50.7 |
| | BT | **77.7** | **56.0** |

### B.2 ANALYSIS OF DATA FILTERING

In Section 3.2, we introduced a filtering step for the data constructed. Here we conduct experiments to analyze the necessity of this step. Specifically, we perform an ablation study on the filtering process using the Qwen-2.5-0.5B model (Qwen Team, 2024), comparing the model's performance before and after applying filtering. The results in Table 6 demonstrate that filtering improves the model's performance. We have presented examples of noise introduced by the absence of filtering in Figure 5, further highlighting the necessity of this step.

Table 6: Ablation study of data filtering.

| Method | Acc | Acc-OOD |
|---|---|---|
| Before filtering | 76.2 | 54.9 |
| After filtering | **76.5** | **55.2** |

The choice of $\tau$ represents a trade-off: setting it too high introduces excessive noisy data, while setting it too low reduces the number of training samples. To provide insight into the selection of $\tau$, we analyze the similarity distribution of positive and negative samples before data filtering, as shown in Figure 7. The results reveal a bimodal distribution. Higher similarity values indicate a greater likelihood of noisy samples, leading us to hypothesize that one of the components corresponds to noise. Using Gaussian Mixture Model (GMM) analysis, we identify an approximate threshold of 0.87 to distinguish noisy samples. Consequently, the selection of 0.85 for $\tau$ in the main text is considered reasonable.

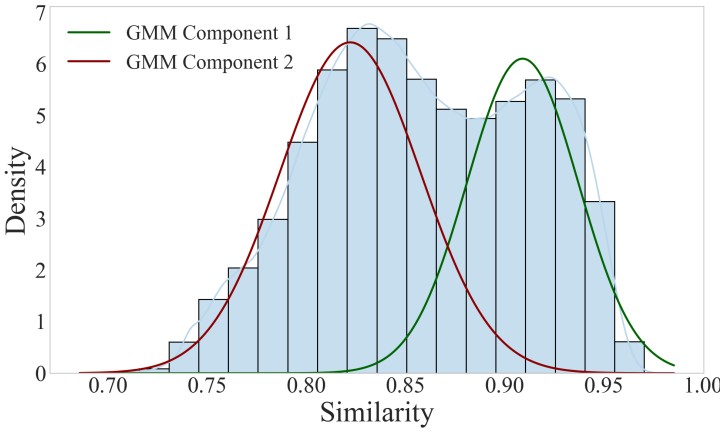

Figure 7: The distribution of similarities between positive and negative samples.

## C EXAMPLE SOLUTIONS

We show several typical theorems proven by CARTS below.

```
Case 1: mathd_algebra_148

theorem mathd_algebra_148 (c : ℝ) (f : ℝ→ℝ)
(h₀ : ∀x, f x =c *x^3 − 9 *x +3)
(h₁ : f 2 =9) : c =3 :=
by
    rcases eq_or_ne c 1 with hc | hc
    cases c
    all_goals simp_all [h₀]
    on_goal 1 => norm_num at h₁
    linarith only [h₀, h₁]
```

Figure 8: **Case 1.** Proved theorem `mathd_algebra_148` by CARTS in mini-F2F-test of ReProver-Lean 4 (Yang et al., 2023). This theorem is proved in 68 expansions.

```
Case 2: mathd_numbertheory_234

theorem mathd_numbertheory_234
  (a b : ℕ)
  (h₀ : 1 ≤a ∧a ≤9 ∧b ≤9)
  (h₁ : (10 *a +b)^3 =912673) :
  a +b =16 :=
by
    simp only [Nat.one_le_iff_ne_zero] at h₀
    obtain ⟨h₂, h₃, h₄⟩ :=h₀
    interval_cases a <;> interval_cases b <;> simp_all
```

Figure 9: **Case 2.** Proved theorem `mathd_numbertheory_234` by CARTS in mini-F2F-test of InternLM2-Plus-1.8B (Ying et al., 2024b). This theorem is proved in 12 expansions.

Case 3: `mathd_numbertheory_135`

```
theorem mathd_numbertheory_135
  (n A B C : ℕ)
  (h₀ : n =3^17 +3^10)
  (h₁ : 11 | (n +1))
  (h₂ : [A,B,C].Pairwise (·≠·))
  (h₃ : {A,B,C} ⊆Finset.Icc 0 9)
  (h₄ : Odd A ∧Odd C)
  (h₅ : ¬3 | B)
  (h₆ : Nat.digits 10 n =[B,A,B,C,C,A,C,B,A]) :
  100 *A +10 *B +C =129 :=
by
    rw [h₀] at h₁
    simp [h₀] at h₆
    linarith [h₆]
```

Figure 10: **Case 3.** Proved theorem `mathd_numbertheory_135` by CARTS in mini-F2F-test of InternLM2-Plus-1.8B (Yang et al., 2023). This theorem is proved in 38 expansions.

Case 4: `mathd_numbertheory_314`

```
theorem mathd_numbertheory_314
  (r n : ℕ)
  (h₀ : r =1342 % 13)
  (h₁ : 0 < n)
  (h₂ : 1342 | n)
  (h₃ : n % 13 < r) :
  6710 ≤n :=
by
    obtain ⟨k, hk⟩ :=h₂
    contrapose! h₁
    rw [hk] at h₁
    have h₂ : k < 5 :=by linarith
    interval_cases k <;> simp_all
```

Figure 11: **Case 4.** Proved theorem `mathd_numbertheory_314` by CARTS in mini-F2F-test of StepProver (Wu et al., 2024). This theorem is proved in 159 expansions.

Case 5: `artin_exercise_13_6_10`

```
theorem exercise_13_6_10 {K : Type*} [Field K] [Fintype Kˣ] :
  (Π x : Kˣ, x) =-1 :=
by
    have h : Πx : Kˣ, x =Πx : Kˣ, x^(-1-1) :=by congr
    simp only [inv_inv] at h
    haveI : DecidableEq K :=Classical.decEq K
    apply FiniteField.prod_univ_units_id_eq_neg_one
```

Figure 12: **Case 5.** Proved theorem `artin_exercise_13_6_10` by CARTS in ProofNet of Step-Prover (Wu et al., 2024). This theorem is proved in 119 expansions.

