# OpenReview forum: "CARTS: Advancing Neural Theorem Proving with Diversified Tactic Calibration and Bias-Resistant Tree Search"
_ICLR.cc/2025/Conference — ICLR 2025 Poster_

### Official Review · Reviewer_YALN · 2024-11-01

**Soundness:** 2
**Presentation:** 4
**Contribution:** 4
**Rating:** 6
**Confidence:** 4

**Summary:**

This paper designs a new search algorithm for formal theorem proving called CARTS. Compared with the standard MCTS algorithm, CARTS encourages diverse tactics that lead to significantly different next states. This paper also trains a value function using pairwise feedback. When applied to three different models, this paper shows that CARTS outperforms baseline methods such as best-first search and MCTS on miniF2F-test and ProofNet at the test time (in other words, the model that generates the tactic is not trained).

**Strengths:**

-	The algorithm, CARTS, introduces a more aggressive exploration strategy compared with the standard MCTS algorithm, by encouraging the diversity of the next state using the MMR algorithm. While the idea of encouraging the diversity of next states is explored in prior works such as [1], using an embedding model to measure the diversity of the next state is novel.
-	This paper identifies the label imbalance problem when training the value function, and mitigate this issue by training the value function using pairwise feedbacks.

[1] Xin, Huajian, et al. "DeepSeek-Prover-V1. 5: Harnessing Proof Assistant Feedback for Reinforcement Learning and Monte-Carlo Tree Search." arXiv preprint arXiv:2408.08152 (2024).

**Weaknesses:**

My major concern is the statistical significance of the results. On the one hand, the CARTS algorithm in this paper has many new components compared with the standard MCTS algorithm, and the design of the new components seems arbitrary (or at least heavily relies on vague heuristics). On the other hand, the improvement of CARTS on miniF2F-test is less than 2%. Since the miniF2F test set only has 244 questions, it means that the difference only comes from a few questions, which is not significant enough given the complexity of the algorithm design.

A few claims in the paper are not justified by empirical evidence:

-	Line 86: “This could result in label imbalance, causing the cross-entropy loss used in training, the value function to easily converge to local optima.” Why does label imbalance with cross-entropy loss lead to local optima?
-	Line 287: I don’t see how the adjustment term can mitigate the domain gap between the training and test datasets. The following sentences in this paragraph are irrelevant to the domain gap.

**Questions:**

-	How good is the off-the-shelf sentence embedding model when applied to Lean’s states? Since the standard sentence embedding model is trained on natural language data, it is unclear to me whether they can distinguish the subtle differences in Lean’s states. Can you show some concrete examples to demonstrate the performance of $f_{enc}$?
-	Is there a typo in Eq (1)? Currently, the bonus term might be bigger than 1 if the action a is only taken a few times.
-	In Table 3, what is the algorithm for the “Bias-resistant value function” row? Is the value function the only difference to CARTS, and how is the alternative value function trained?
-	The choice of the Bradley-Terry model sounds arbitrary, although the argument that the BT model effectively oversamples the positive pairs makes sense. Is the value function trained with BT model better than the value function trained with binary cross entropy loss with explicit oversampling of positive examples?
-	Can w(s,a), or MMR(s,a), be negative? If w(s,a) is negative, the exploration bonus (the second term in Eq. 1) is then increasing when action a is taken more times.
-	In the ablation for k, the conclusion is “as k increases, the pass@1 value generally decreases”. What about even smaller k such as k = 1,2,4? Intuitively the performance when k=1 should be very bad.
-	The case study in Section 4.3 shows that the ranking of the right tactic is improved. This could be a good place for more ablation studies. Which feature of the algorithm contributes to this improvement, length penalty, MMR, the new value function, or the adjustment term?

---

> ### Author Response · Authors · 2024-11-22
> **Rebuttal by Authors (Part 1)**
>
> Thank you for taking the time to review our paper and provide feedback! Below we address your questions and concerns.
>
> **1. About statistical significance**
>
> We would like to clarify that the improvement achieved by our method occurs *without requiring additional data* to train the tactic generator. Notably, Deepseek-Prover [1] achieved an 11.4% improvement only after leveraging an additional dataset 13 times the size of Mathlib (3.108B vs. 0.238B) (see [1], Table 2), highlighting the inherent difficulty of the theorem proving task. We argue that the consistent improvement of our method (about 2%) across different pre-trained tactic generators can demonstrate the effectiveness of our approach.
>
> **2. About Label imbalance**
>
> The term "local optima" refers to that CE loss training tends to prioritize predicting the majority class when faced with label imbalance, resulting in suboptimal performance. About empirical evidence, we provide additional experiments comparing the performance of CE and BT in Appendix B.1. The results demonstrate that BT consistently outperforms CE in situations involving class imbalance. Further details of the experiments can be found in our revised version.
>
> **3. About domain gap**
>
> The reward adjustment term can be interpreted as a form of test-time adaptation to the distribution of test data. If the number of valid tactics is low, the value of should be decreased, as the current state is more likely to correspond to a wrong state. Conversely, if the number of valid tactics is high, the value should be increased to encourage exploration. As a result, the adjustment term can mitigate the domain gap. We make corresponding revisions to the content in the main text (Line 303). About empirical evidence, we conduct additional experiments in Appendix B.1 to demonstrate the severity of the domain gap. Furthermore, the ablation study presented in Table 3 has shown the effectiveness of the adjustment term.
>
> **4. Example of $f_{enc}$**
> The sentence embedding model demonstrates a relatively strong capability for recognizing similarities. Below is an example to illustrate this point (MiniF2F, amc12_2000_p6), where two tactics, ‘intro h’ and ‘intro H’, yield the following next states:
> ```
> p q : ℕ
> h₀ : Nat.Prime p ∧ Nat.Prime q
> h₁ : 4 ≤ p ∧ p ≤ 18
> h₂ : 4 ≤ q ∧ q ≤ 18
> h:  ¬p * q - (p + q) = 194
> ⊢ false
> ```
> ```
> p q : ℕ
> h₀ : Nat.Prime p ∧ Nat.Prime q
> h₁ : 4 ≤ p ∧ p ≤ 18
> h₂ : 4 ≤ q ∧ q ≤ 18
> H:  ¬p * q - (p + q) = 194
> ⊢ false
> ```
> The similarity score calculated using the pretrained sentence embedding model is 0.998, indicating that the two states are recognized as similar. Then our diversified tactic calibration can be applied for deduplication.
>
>
>
> **5. About typo in Eq (1)**
>
> We would like to clarify that there is no typo in Eq. (1). The expression for the WUCT score in Eq. (1) is consistent with the formulation of PUCT presented in DT-Solver [2]. The bonus term is divided by the visit count $N(s,a)$, ensuring that its value does not exceed 1.

---

> ### Author Response · Authors · 2024-11-22
> **Rebuttal by Authors (Part 2)**
>
> **6. Clarify algorithms in Table 3**
>
> We would like to clarify that in Table 3, the algorithm labeled “Bias-resistant Value Function” refers exclusively to our proposed bias-resistant value function integrated within the MCTS framework. In contrast, the label “Diversified Tactic Calibration” denotes the MCTS framework incorporating only the diversified tactic calibration. In this case, the value function does not require training and instead directly utilizes the intrinsic reward, as described in Line 367.
>
> **7. Effectiveness of BT model**
>
> Thank you for your suggestion.
> We have included a comparative experiment between the BT model and the binary cross entropy loss (CE) in Appendix B.1. Using Qwen2.5-0.5B [3] as an example, the results are summarized as follows:
> | Method            | Acc (%) |
> |--------------------|---------|
> | CE                | 63.2    |
> | CE (oversampling) | 67.6    |
> | BT                | **76.5**    |
>
> We find that the BT model consistently outperforms CE in our experiments. While the use of oversampling improves the performance of CE, it still does not surpass the BT model. Additional experiments on other models and further experimental details can be found in Appendix B.1 of the revised version.
>
> **8. Can $w(s,a)$ be negative?**
>
> The value of $w(s,a)$ cannot be negative. Therefore, in the actual implementation, we set $w(s,a) =\max (0,\text{MMR}(s,a))$ to ensure that all $w$ are non-negative (clarified in Line 241). Additionally, this issue can be avoided by setting a large value for $\lambda$.
>
>
> **9. Ablation study on $k$**
>
> We supplement the ablation study results for $k = 1, 2, 4$, as shown in the table below. Additionally, the corresponding results are incorporated into Figure 3 (only the $k = 4$ is included for aesthetic purposes).
>
> | $k$       | 1    | 2    | 4    | 8| 32| 64|
> |-------------|------|------|------|------|------|------|
> | Pass@1 (\%) | 30.3 | 32.0 | 33.6 | **37.7** |37.3 | 36.5|
>
> The experimental results demonstrate that as $k$ increases, the pass@1 initially improves, reaching a peak, and then gradually declines. Therefore, we recommend choosing a moderate $k$, such as $k=8$, in practice.
>
>
> **10. Analysis for the ranking of the right tactics**
>
> Thank you for your advice. The MMR in diversified tactic calibration influences the ranking of the $w$ values for the correct tactics. Using Reprover-Lean4 on miniF2F (for k = 8, E = 64), we statistically analyze the improvement in the ranking of correct tactics on the first state of proved theorems. We find that diversified tactic calibration can lead to an average ranking improvement of 14.8 for the right tactics. The bias-resistant value function has no effect on the ranks because it does not modify $w$.
>
>
> Reference:
>
> [1] Xin, Huajian, et al. DeepSeek-Prover: Advancing Theorem Proving in LLMs through Large-Scale Synthetic Data.
>
> [2] Wang, Haiming, et al. DT-Solver: Automated Theorem Proving with Dynamic-Tree Sampling Guided by Proof-level Value Function.
>
> [3] Qwen Team. Qwen2.5: A party of foundation models.

---

> ### Comment · Reviewer_YALN · 2024-11-23
>
> Thank you for the detailed response. I’ve raised the score accordingly.
>
> Regarding Eq (1), if I understand correctly, if the state s has two actions a_1 and a_2, then N(s,\cdot) = N(s,a_1) + N(s,a_2). Isn’t it possible that N(s,a_1)=100, N(s,a_2)=1 so that the term \sqrt{N(s,\cdot)}/N(s,a_2) is bigger than 1? Could you clarify?

---

> > ### Author Response · Authors · 2024-11-23
> >
> > We sincerely apologize for our earlier misunderstanding of your question. Regarding the second term in Eq. (1), it is indeed possible for this term to exceed 1. As in the example you provided, if $a_1$ has been explored many times, $N(s, a_1)$ will be very large. In this case, $\sqrt{N(s, \cdot)}/N(s, a_2)$ can be greater than 1. However, this is not an issue; in fact, it is precisely because this term becomes large that $a_2$ will subsequently be explored, thereby fostering broader exploration as intended. Our Eq. (1) is also similar to those in [1, 2].
> >
> > Finally, we greatly appreciate your valuable feedback on this work.
> >
> > [1] Guillaume Lample et al. Hypertree Proof Search for Neural Theorem Proving. NeurIPS 2022.
> > [2] David Silver et al. A General Reinforcement Learning Algorithm That Masters Chess, Shogi, and Go Through Self-Play. Science, 2018.

---

### Official Review · Reviewer_JC1S · 2024-11-02

**Soundness:** 4
**Presentation:** 4
**Contribution:** 4
**Rating:** 8
**Confidence:** 2

**Summary:**

This work proposed two ways to enhance neural theorem proving performance. One is to caliberate generated tactic, which can alleviate the problem of tactic redundancy and further helps to diversify the proof path. The second is to use bias-resistant value function by introducing preference modeling and adjustment according to valid tactic ratio. Experimental results show the combination of this two components could achieve remarkable performance gain.

**Strengths:**

1. The presentation of this paper is very clear, vitual demonstration is easy to understand as well.
2. The proposed method is reasonable and well-motivated, the first component is motivated by the redunancy of generated tactic and the second component is motivated by common gap-bias between different dataset.
3. The reported results is good and stable, it can be applied to many existing methods and achieve a boost in performance.

**Weaknesses:**

1. The proposed method is not that generalizable for now, as it can be only applied to one-step tree search methods.
2. The operating logit looks a little bit complex, raising concern about the running effiency, for example, in the tactic calibration process, this method need to run the tactic to get the next state at first and then perform MMR algorithm, which could be time-consuming.

**Questions:**

1. What's the comparison beween this method and other related method in terms of running efficiency? It would be great to clarify this point.

---

> ### Author Response · Authors · 2024-11-22
> **Rebuttal by Authors**
>
> Thank you for your valuable feedback.  Below are our responses to your questions:
>
> **About generalizability:** Our method is specifically designed for one-step tree search approaches, which currently represent the mainstream in ATP.  We are also exploring the possibility of extending our method to multi-step tree search approaches in our future work.
>
> **About computational efficiency:** The additional computational cost of our method compared to baseline methods (BFS and MCTS) primarily comprises two components: obtaining embeddings and executing the MMR algorithm.  We utilize a lightweight embedding model, which keeps the computational cost relatively low.  Regarding the MMR algorithm, the number of tactics expanded is quite limited (typically 64), rendering the computational time for this component negligible.

---

> > ### Comment · Reviewer_JC1S · 2024-11-23
> >
> > Thanks to authors for further clarification. I still believe this is a good work.

---

> > > ### Author Response · Authors · 2024-11-23
> > >
> > > Thank you, we greatly appreciate your valuable advice on this work.

---

### Official Review · Reviewer_cJwa · 2024-11-04

**Soundness:** 3
**Presentation:** 4
**Contribution:** 3
**Rating:** 6
**Confidence:** 4

**Summary:**

This paper introduces two major improvement over the MCTS-based theorem proving framework: (1) diversified tactic calibration which re-scores generated tactics, by the [MMR algorithm](https://www.cs.cmu.edu/~jgc/publication/The_Use_MMR_Diversity_Based_LTMIR_1998.pdf) for diversity-based re-ranking; and (2) bias-resistant value function, with embedding-based tactic filtering and preference modeling with the Bradley-Terry model. Take Reprover-Lean4 (a 0.3B model) as an example, the proposed method CARTS reaches 37.7%, while its MCTS counterpart is 36.5%.

**Strengths:**

- **Impact**: The paper is well-motivated -- improving the efficiency of tactic search and the accuracy of the value function addresses key challenges that practitioners in this field are eager to solve.
- **Originality**: Both designs of tactic calibration and bias-resistant value function are new and useful (incremental) contribution to the existing theorem proving framework.
- **Writing**: The paper is very well-written and easy-to-follow.
- **Practicality**: The proposed method looks to be easy-to-implement, and authors have provided relevant codes. It would be nice if this could be made as a modular design that can be plugged into different pipelines/repositories.

**Weaknesses:**

My concerns are mostly on experiments.
- Both designs are non-trivial, however, it looks like the improvement over the MCTS benchmark is rather marginal. How many independent runs were used? Could the authors report the confidence interval, or provide some discussions on this?
- CARTS is based on MCTS (is this correct?); i wonder if it could be applied on top of BFS as well. Also, what is the value function used for BFS and MCTS in the tables? Do they use the model likelihood or some other value models?
- In Line 267, how to determine $\tau$? Could the authors do some ablations on this?
- Could the authors evaluate the method on more benchmarks, e.g., LeanDojo (i.e., mathlib theorems)?
- The paper uses the phi-1.5 model as the base model for the value function. There has been controversy or potential ethical concerns on this model (e.g., Schaeffer, Rylan. "[Pretraining on the test set is all you need](https://arxiv.org/pdf/2309.08632)." arXiv 2023), though not directly relevant to theorem proving tasks. Could the authors do some ablations with other models, like llama-3.2-3b? Orthogonally, it might be nice to check if the performance will improve as the value model scales up. By the way, please add the model parameter count to reprover and stepprover in the tables.

Overall, i think the paper is well-written and brings useful information to the community, and I lean towards acceptance. I am happy to raise my score if the authors can further address the above concerns.

**Questions:**

Please see the weakness part.

---

> ### Author Response · Authors · 2024-11-22
> **Rebuttal by Authors**
>
> Thank you for taking the time to review our paper and provide feedback! Below we address your questions and concerns.
>
> **1. About independent runs**
>
> Due to limitations in computational resources, we did not conduct multiple experimental runs.  However, we believe the results remain robust, as our proposed method does not introduce additional randomness.  The only source of randomness arises from the LLM’s sampling process, which tends to stabilize with a large sample size.  Prior works [1, 2] have similarly opted not to perform repeated experiments.  Furthermore, theorem proving is a relatively special task—once a theorem is successfully proved in a single run, it is considered solved.
>
> **2. Applicability of CARTS to BFS**
>
> CARTS could indeed be adapted to use a BFS framework. However, we think that MCTS is more suitable for ATP tasks. Our work partly focuses on a biased value function, and BFS is generally less robust in the presence of noise. In contrast, MCTS dynamically adjusts value estimates during exploration, enabling it to better handle noises.
>
> **3. Details of BFS and MCTS in experiemnets**
>
> In our experiments, BFS employs the model likelihood approach commonly used in ATP studies and MCTS utilizes the intrinsic reward, as described in Line 367 in the paper.
>
> **4. Determining $\tau$ and the ablation study**
>
> We provide additional analysis on our data filtering in Appendix B.2. Observing the bimodal distribution of similarity scores (Figure 7), we find that the value of $\tau$ can be determined using a Gaussian Mixture Model (GMM). Furthermore, through the ablation study (Table 6), we demonstrate that omitting filtering introduces noise, which negatively impacts the value function. Detailed results are available in the revised version.
>
>
> **5. Regarding the LeanDojo Mathlib Benchmark**
>
> We did not conduct experiments on the LeanDojo Mathlib benchmark in our study. The primary reason is that some tactic generators (e.g. StepProver) were trained on Mathlib, and our bias-resistant value function was also trained on Mathlib. This overlap introduces a potential risk of data leakage, which could compromise the fairness of comparisons.
> Instead, we opted for experiments on MiniF2F and ProofNet, adhering to the experimental setups established by prior work [2]. We believe that the ATP field would greatly benefit from the development of more standardized benchmarks to ensure consistent and fair evaluations across studies.
>
>
> **5.   Additional experiments of training the value function.**
>
> In Appendix B.1, we present additional experiments to demonstrate the effectiveness of training the value function using the BT modeling approach. We conducted training across three models (Qwen2.5-0.5B, Llama3.2-1B, Llama3.2-3B), with the results summarized in Table 5. Here we show the results for Llama3.2-3B and more results please see our revised version.
>
> | Method            | Acc (%) | Acc-OOD (%) |
> |--------------------|---------|---------|
> | CE                | 63.0 |50.7|
> | BT                | **77.7**|**56.0**|
>
> We find that a slight performance improvement as model size increases;   however, the improvements remain relatively modest.   We attribute this to the limited dataset size, which constrains the emergence of distinct scaling properties.
>
> [1] Yang, Kaiyu, et al. LeanDojo: Theorem Proving with Retrieval-Augmented Language Models.
>
> [2] Wu, Zijian, et al. LEAN-GitHub: Compiling GitHub LEAN repositories for a versatile LEAN prover.

---

> > ### Comment · Reviewer_cJwa · 2024-11-24
> >
> > Thanks for the authors' feedback. I still have concerns about the experiments and will attempt to review the provided code to evaluate them further.

---

### Official Review · Reviewer_kceZ · 2024-11-04

**Soundness:** 3
**Presentation:** 4
**Contribution:** 3
**Rating:** 8
**Confidence:** 3

**Summary:**

The paper propose to improve MCTS for Neural Theorem Proving. It adds a measure of diversity in the exploration that takes into account the diversity of the tactics searched.

**Strengths:**

The paper addresses a hot and interesting topic.
The experimental results show better results than alternative search algorithms.
The experiments are well illustrated

**Weaknesses:**

Only one step tree search is supported.
Lack of experiments with more processing power.

**Questions:**

Does your method scale with more budget?

---

> ### Author Response · Authors · 2024-11-22
> **Rebuttal by Authors**
>
> Thank you for your valuable feedback. Below we address your concerns.
>
> **About experiments:** In Appendix B, we add additional experiments to further demonstrate the effectiveness of our bias-resistant value function.   The main conclusion of the supplement is elaborated in our overall response.
>
> **About the more budget:** We have conducted relevant experiments, as shown in Figure 2 in the main text, to illustrate that our method achieves improvements as the budget scales up.  Experiments with larger budgets may exceed our current computational resource capacity.  However, we believe that the current experiments are sufficient to demonstrate that our method can scale effectively, particularly as the search depth increases.

---

### Author Response · Authors · 2024-11-22
**Overall Response**

We would like to express our sincere gratitude to all the reviewers for their valuable time and constructive feedback on our work.

In response to the reviewers’ comments, we have conducted additional experiments to demonstrate the effectiveness of our bias-resistant value function. These experiments are presented in detail in Appendix B. From these supplementary analyses, we draw the following conclusions:

1. The Bradley-Terry (BT) model effectively addresses the issue of class imbalance in training the value function.
2.Our analysis underscores the critical importance of data filtering and provides a way for determining the hyper-parameter $\tau$.

Further details are available in the revised version of the paper.

We greatly appreciate your insightful suggestions and look forward to engaging in further discussions.

---

### Meta-Review · Area_Chair_VKgb · 2024-12-21

**Metareview:**

This paper proposes CARTS, a novel framework for neural theorem proving that introduces two key innovations: Diversified Tactic Calibration, which uses the MMR algorithm to enhance tactic diversity, and a Bias-Resistant Value Function, leveraging the Bradley-Terry model to address label imbalance. The reviewers unanimously appreciated the contributions, practical relevance, and consistent improvements over baselines. The method's ability to address critical challenges, such as tactic redundancy and bias in value estimation, makes it a meaningful contribution to the field.

The rebuttal was thorough and addressed many of the reviewers’ concerns, including scalability, statistical significance, and experimental clarity. The authors provided additional experiments and detailed responses that strengthened the paper's overall presentation and impact. I agree with the reviewers that this work is a valuable addition to the neural theorem proving community.

That said, some concerns remain, such as the limited evaluations on larger benchmarks (e.g., LeanDojo), modest performance gains (~2% improvement on miniF2F-test), and the method’s focus on one-step tree search. These points are important and should be addressed in future work.

**Additional Comments On Reviewer Discussion:**

Overall, the paper is well-written, impactful, and makes novel contributions to the field. The authors are encouraged to incorporate feedback from the reviews and rebuttal, particularly by expanding empirical evaluations and exploring broader applicability, in future revisions.

---

### Decision · Program_Chairs · 2025-01-22

Accept (Poster)